# Domain-PFP allows protein function prediction using function-aware domain embedding representations

Nabil Ibtehaz [1], Yuki Kagaya [2] & Daisuke Kihara [1,2]✉

Domains are functional and structural units of proteins that govern various biological functions performed by the proteins. Therefore, the characterization of domains in a protein can serve as a proper functional representation of proteins. Here, we employ a self-supervised protocol to derive functionally consistent representations for domains by learning domain-Gene Ontology (GO) co-occurrences and associations. The domain embeddings we constructed turned out to be effective in performing actual function prediction tasks. Extensive evaluations showed that protein representations using the domain embeddings are superior to those of large-scale protein language models in GO prediction tasks. Moreover, the new function prediction method built on the domain embeddings, named Domain-PFP, substantially outperformed the state-of-the-art function predictors. Additionally, Domain-PFP demonstrated competitive performance in the CAFA3 evaluation, achieving overall the best performance among the top teams that participated in the assessment.

[1] Department of Computer Science, Purdue University, West Lafayette, IN, USA. [2] Department of Biological Sciences, Purdue University, West Lafayette, IN, USA. ✉email: dkihara@purdue.edu

Protein function prediction is one of the long-standing, fundamental topics of bioinformatics, which involves profiling the activities and interactions of proteins[1]. Although protein functions are eventually determined by experiments, the experimental effort and expense slow down the process of function discovery, which is in contrast to the ever-increasing volume of sequenced proteins[2]. At present, not even 1% of sequenced proteins have functional annotation[3]. Unlike relatively cheaper sequencing technologies, there is a deficit of scalable, high-throughput experimental assays to functionally annotate proteins[4]. This has led to the demand for in-silico methods of automated protein function prediction[5]. Protein functions have been determined naturally from sequence similarity to known proteins[6] and other characteristics of proteins that can trace functional relevance. Such information includes structural configuration[7–9], phylogenetic information[10,11], domain distribution[12–14], protein networks[3,15], and combinations of multiple sources[16,17]. Recently, various deep learning-based methods were proposed to learn a functional representation of proteins[8,16,18–22]. Such methods demonstrated substantial improvement over traditional database search-based methods[23,24].

Proteins consist of domains, which are functional and structural units responsible for specific functions and interactions[25]. Therefore, it is compelling to infer the functions of a protein based on the presence and distribution of the various domains in it. InterPro2GO is an ongoing project that assigns GO annotations to specific domains in the InterPro database, and this annotation is done manually by experts[26,27]. Although the domain-GO mapping by InterPro2GO provides curated information on protein function, the coverage is severely limited. For example, there are approximately 38k InterPro entries and 48k GO terms, but the current version of InterPro2GO (version-date: 2022/03/16) mapping only includes 16,443 unique InterPro entries and 6,482 GO terms. Despite the lack of annotations, several methods have tried to leverage protein domain information for function prediction. Messih et al. analyzed the recurrence and order of protein domains and their influence on protein functions[13]. Rojano et al. attempted to associate domains and functions through tripartite graphs[14]. Besides such domain-focused studies, protein domains have been consistently used as a source of complementary functional information in a number of ensemble methods[3,16,17], and some analyses even revealed that domain information is the most crucial one[16].

As in many other areas in bioinformatics, deep learning has been applied for function prediction from domain information. However, the effective use of domains is critically constrained by low coverage of functional assignments, high dimensionality, and acute data imbalance. For instance, in a recent competitive deep-learning-based model, DeepGOZero[22], a 26,406-dimensional input of InterPro feature vectors was reduced to 1024 dimensions using a single multi-layer perceptron (MLP) layer, which results in considerable information loss. A similar situation is observed in DeepGraphGO[21] as well.

Here, we introduce Domain-PFP, a protein function prediction method that uses functional representation of proteins through domain-GO association learned by a self-supervised method from protein databases. Self-supervised learning is based on the idea of leveraging the inherent co-occurrence relationship of complementary information in the data to learn new labels in a semi-automatic process[28]. We used self-supervised learning because it can directly learn domain and GO co-occurrence from abundant protein sequences and is able to alleviate the problem of current domain databases, where many domains do not have function annotation. Following the underlying concepts of self-supervised learning, we first learned pseudo-labels of GO prediction probability from individual domain terms. Then, we derived the dense representation of domains consistent with functional information to characterize protein sequences and used the representation to predict protein functions. The embeddings learned both at the domain and protein level have turned out to be functionally meaningful as the embedding distance showed substantial negative correlations with functional similarity[29] of GO terms that are present in the domains and protein sequences. Moreover, a systematic comparison with large-scale Protein Language Model (PLM) representations[30,31], which use variants of Transformers[32] and BERT[33] architectures, and have demonstrated success in function prediction[34,35], revealed that our embeddings are more applicable for function prediction, despite being a fraction of the aforementioned PLM complexity. This improvement is further vividly observed in challenging cases of predicting rare and more specific functions. In addition, using a straightforward K-Nearest Neighbors (KNN) model with the learned embeddings along with sequence similarity and interaction information, Domain-PFP remarkably outperforms more complex state-of-the-art methods. Most notably, Domain-PFP achieved an increase in the area under precision-recall curve (AUPR) by 2.43%, 14.58%, and 9.57% over the state-of-the-art method for molecular function (MF), biological process (BP), and cellular components (CC), respectively. Domain-PFP has also demonstrated competitive performance when compared with top-scoring methods in the CAFA3 evaluation[36].

## Results

**Dataset of domains and GO annotations.** We collected 568,002 protein sequences from Swiss-Prot (release 2022_3)[37] and assigned InterPro domains using InterProScan 5 REST API[38]. Despite InterPro maximizing domain coverage by combining entries from 13 databases, 36,403 proteins had no InterPro annotations, so we discarded them. Concurrently, we collected GO terms for protein sequences from UniProt. We considered both experimentally and computationally assigned functions since IEA (Inferred from Electronic Annotation) terms demonstrated increased accuracy in our previous works[6]. We also propagated the parent GO terms using the core ontology release 2021-01-01. In summary, our dataset contained 531,599 proteins with 32,471 unique domains and 33,199 unique GO terms (8,297, 21,805, and 3,097 MF, BP, and CC terms, respectively).

**Self-supervised learning for domain-GO embeddings.** Using the domain and GO term assignments to protein sequences, we computed the conditional probability of a protein that contains $domain_i$ having the $GO_j$ function:

$$p\left(GO_j | domain_i\right) = \frac{p\left(domain_i \cap GO_j\right)}{p\left(domain_i\right)} \quad (1)$$

Here, $p\left(domain_i\right)$ represents the probability of a protein containing $domain_i$, while $p\left(domain_i \cap GO_j\right)$ represents the joint probability of a protein with $domain_i$ performing $GO_j$. We can calculate both probabilities from the co-occurrence relationships of domains and GO terms in the dataset by counting the occurrences. These probabilities serve as the pseudo labels or target function for our self-supervised learning method.

Our ultimate goal is to predict protein functions. To achieve this, we aim to develop a representation of domains that, in conjunction with a learned representation of GO terms, is consistent with the domain-GO co-occurrence conditional probability. In other words, we seek to design two representations or embeddings, $\phi$ and $\psi$, which separately represent domains and GO terms, respectively, and a bivariate function $f$ that can map

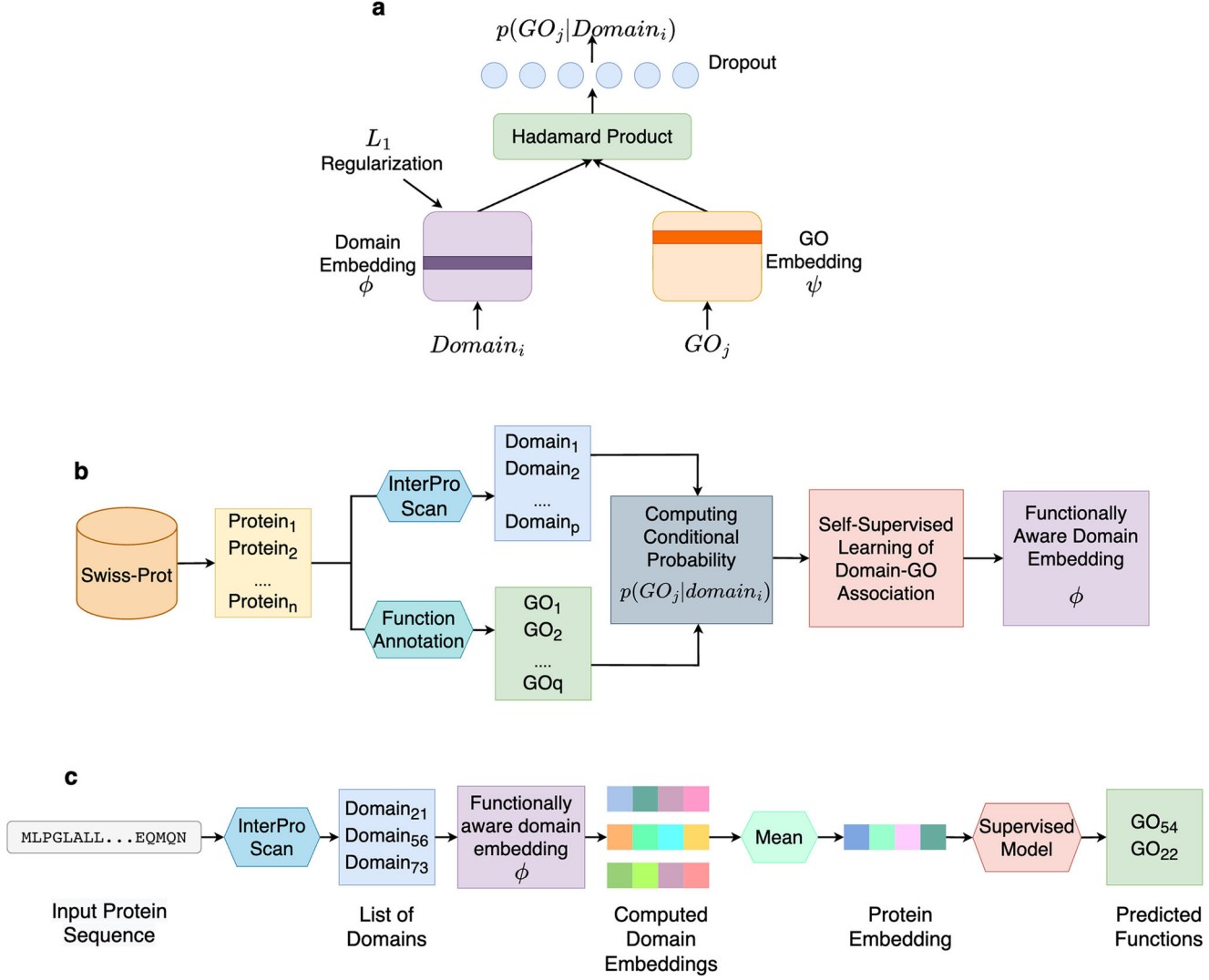

**Fig. 1 Overview of Domain-PFP. a** The network architecture used for self-supervised learning of domain embeddings. **b** The overall pipeline of learning the functionally aware domain embeddings. **c** The steps of computing the embeddings of a protein and inferring the functions.

the conditional probability of the co-occurrence of any $domain_i$ and $GO_j$:

$$f\left(\phi\left(domain_i\right), \psi\left(GO_j\right)\right) \rightarrow p\left(GO_j|domain_i\right) \quad (2)$$

In our case, we utilized two 256-dimensional embedding matrices $\phi$ and $\psi$, as representations for domains and GO terms, respectively. The bivariate function $f$ was modeled as a two-layer densely connected network that takes the Hadamard product of $\phi(domain_i)$ and $\psi(GO_j)$ as input, decomposes the values in a 128-dimensional space, and finally predicts the conditional probability $p(GO_j|domain_i)$. The network architecture is presented in Fig. 1a, where the function $f$ is represented as an array of circles in light blue. Concretely, $f$ takes the following form:

$$f = W_2(RELU(W_1\left(\phi\left(domain_i\right) \odot \psi\left(GO_j\right)\right) + b_1)) + b_2 \quad (3)$$

where $(W_1, b_1)$ and $(W_2, b_2)$ are weights and biases from the first and the second layer of the network, respectively. The Hadamard product of the two embeddings is represented by the symbol $\odot$. The network is regularized by dropout, and the domain embedding matrix $\phi$ is further regularized by L1-norm to impose sparsity. The $\phi$ embedding matrix for each domain, as well as the

$\psi$ embedding for each GO term, were learned through back-propagation with the mean squared error (MSE) loss using the Adam optimizer with default settings[39]. We intended to keep the function $f$ simple so that the domain embeddings could effectively learn functional relevance, rather than letting the function $f$ learn the correlation between domain and GO term co-occurrence. This is inspired by a recent work, which demon-strated that a strong encoder in conjunction with weak decoder results in a strong representation learner[40]. The function $f$ provides the association probability between a domain and a GO term (Eq. 1), which we name DomainGO-prob. We trained three different versions of DomainGO-prob for the three sub-ontologies, MF, BP and CC, respectively.

The overall pipeline for learning the domain embeddings is summarized in Fig. 1b. We started by collecting annotated protein sequences from Swiss-Prot, along with domain and GO term assignments. Domains were obtained from InterProScan, while GO terms were collected from Swiss-Prot. Next, we calculated the conditional probabilities of all the domain-GO associations by counting their co-occurrences in the dataset. Finally, the domain embeddings ($\phi$) and GO term embeddings ($\psi$) were computed using the network shown in Fig. 1a. The network was trained and validated on the aforementioned dataset

of 32,471 unique domains and 33,199 unique GO terms. We randomly selected 80% of the domain-GO pairs for training and used the remaining 20% for validation. Three different models, i.e., three different sets of embeddings were developed for the three sub-ontologies. The details of the network training are described in the Methods.

**Predicting GO terms for a query protein (Domain-PFP).** Using the computed domain embeddings, we represented a protein, which may be composed of several domains, as the average of the embeddings of all the domains in it. This is similar to how PLM encodes proteins by averaging the individual residue level representations[31]. For a protein $P_k$ with domains $d_{P_k}$ the protein embedding is computed as

$$D(P_k) = \frac{\sum_{d \in d_{P_k}} \phi(d)}{|d_{P_k}|} \quad (4)$$

With the protein embedding, we can use supervised classifiers to infer protein functions. Here, we used a KNN classifier, following the convention of BLAST or PPI network scoring[16,21]. KNN models using protein language models have also been shown to be on par with top methods of Critical Assessment of Functional Annotation 3 (CAFA3)[34]. The confidence score of annotating a protein $p_i$ with the GO term $GO_j$, $S_D(p_i, GO_j)$ is computed as follows:

$$S_D(p_i, GO_j) = \frac{\sum_{p_k \in K_{neigh}} I(p_k, GO_j) \times ||D(p_i) - D(p_k)||^2}{\sum_{p_k \in K_{neigh}} ||D(p_i) - D(p_k)||^2} \quad (5)$$

where $K_{neigh}$ is a neighborhood of $K$ proteins, and $I(p_k, GO_j)$ is 1 if the protein $p_k$ is annotated with $GO_j$, and 0 otherwise.

The steps of computing protein embeddings and predicting functions are outlined in Fig. 1c. For a given query protein sequence, domains are assigned using InterProScan[38], and their individual domain embeddings are obtained. The embedding of the query protein is then computed by taking the average of the assigned domain embeddings (Eq. 4). Finally, the protein embedding is used to find known proteins that are close in the embedding space (Eq. 5) using a supervised classifier (KNN for our approach) to infer its functions.

**Correlation of embedding distance and functional similarity of domains and proteins.** To start with, we analyzed how the distance of the domain embeddings correlates with the functional similarity of domains and proteins. Having functionally similar domains close in the embedding space is essential for the embeddings to be useful for function prediction. As a measure of the embedding distance, we adopted the Manhattan distance as it is discussed to be more meaningful in high-dimensional spaces than, for example, the Euclidean distance[41]. As for functional similarity, we computed the Jaccard Index following a previous work[3]. For a domain, we considered GO terms are assigned to the domain if they have a conditional probability no less than 0.5, i.e., $GO\ Terms = \{GO_i : p(GO_i|domain) \ge 0.5\}$. This set of assigned GO terms for domain A and B are denoted as $GO\ Terms_{domainA}$ and $GO\ Terms_{domainB}$ in the following equation. The Jaccard Index for two domains, A and B is defined as

$$Domain\ Functional\ Similarity(domain_A, domain_B)$$
$$= \frac{|GO\ Terms_{domainA} \cap GO\ Terms_{domainB}|}{|GO\ Terms_{domainA} \cup GO\ Terms_{domainB}|} \quad (6)$$

We randomly selected 100,000 pairs of domains and computed their functional similarity relative to the embedding distance in

Fig. 2a. Domain functional similarity was computed separately for each of the three GO categories. Overall, a negative correlation was observed between the embedding distance and functional similarity for all three GO categories. Substantial Jaccard Index values, such as those over 0.5, were observed mainly for domain embedding pairs that were close in distance, for example, <10. Almost all domain pairs with a large distance, for example, a distance of 20 or higher for MF and CC and over 10 for BP, had a small functional similarity value of <0.2. A perfect Jaccard Index of 1.0 was only observed for domain pairs with a relatively small embedding distance. Thus, it is evident that our model generates similar embeddings for functionally similar domains.

We have also examined protein-level functional similarity relative to the embedding similarity (Fig. 2b). As the measure of the functional similarity of proteins, which are annotated by multiple GO terms in the three categories, we used the *funSim* score[42]. *funSim* essentially computes the average of semantic similarity of best matching GO terms from two proteins for each GO category, and then averages the score over the three GO categories (for the concrete definition, see Methods). *funSim* score ranges from 0 to 1 with 1 as the maximum score.

We took 1,000,000 random pairs of proteins and computed their embeddings for MF, BP, and CC separately and concatenated them to obtain the overall embedding. In Fig. 2b, mean *funSim* score of protein pairs were plotted relative to the Manhattan distance of protein embeddings. We can see the overall trend that *funSim* score drops as protein embeddings become more distant from each other. Large *funSim* scores were observed only for close protein embeddings, e.g., a Manhattan distance of <5.

Overall, in this section, we confirmed that functionally similar domains and proteins are placed close to each other in the embedding space.

**Learning InterPro2GO annotations.** Next, we examined how well our domain embeddings align with expert-curated GO mappings of InterPro2GO. For this analysis, we used the InterPro2GO mapping of version-date 2022/03/16[43], which comprises 35,046 mappings between 16,443 unique InterPro domain entries and 6,482 unique GO terms. We considered 34,832 InterPro2GO annotations, excluding 214 mappings with domains or GO terms that are not included in our dataset.

For all the domain-GO pairs in the InterPro2GO mappings, we predicted the conditional probability using DomainGO-prob (Fig. 1a) that the GO exists in the domain. The results are shown in Fig. 2c (using orange bars). As shown, for over 80% of cases, existing GO term-domain associations have a high score of over 0.9 (the rightmost bar) for all three GO categories. Thus, DomainGO-prob was able to associate GO terms to protein domains using the self-supervised learning protocol that associated GO terms and domains from the co-occurrences in full protein sequences.

We further conducted experiments with an adversarial version of this analysis. Namely, to test the generalization ability to learn from the context of related, co-occurring domains and GO terms alone, we removed all the probability values of domain-GO pairs that exist in the InterPro2GO mapping and then re-trained the DomainGO-prob models. Formally, from the original dataset $\mathcal{D} = \{(domain_i, GO_j)\}$ we constructed a new dataset $\mathcal{D}' = \{(domain_i, GO_j) : (domain_i, GO_j) \notin InterPro2GO\}$. With this dataset $\mathcal{D}'$, we re-trained the DomainGO-prob models and examined the conditional probability of GO terms that exist in InterPro2GO. The results are represented by the blue bars in Fig. 2c. Under this setting, DomainGO-prob predicted a score >0.5 for 66.5%, 81.9%, and 86.5% for MF, BP, and CC,

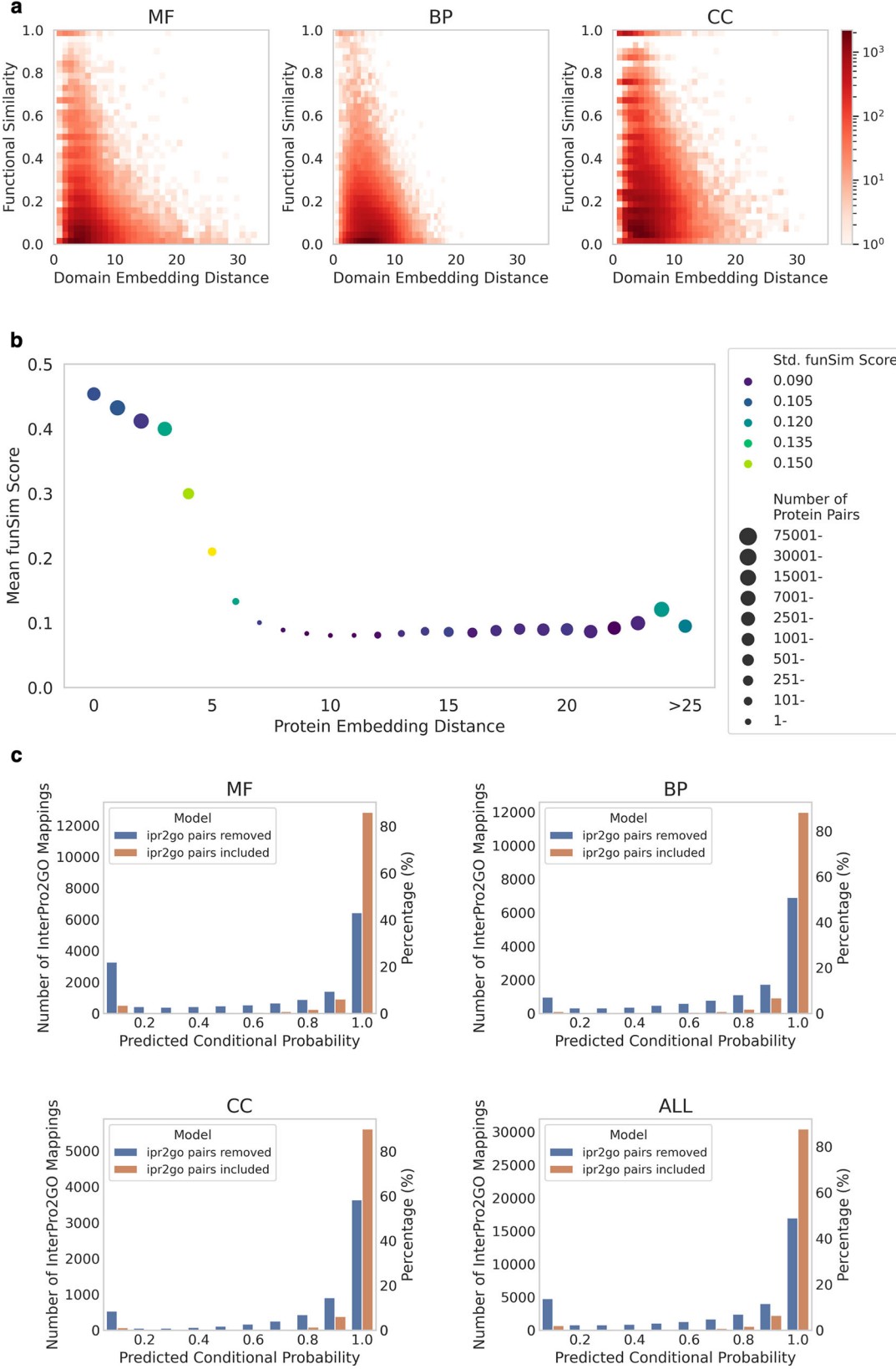

respectively. Thus, even without explicit knowledge, DomainGO-prob was able to extract the meanings of domain-GO relationships only from the contextual information of co-occurrences and hierarchies. Among the three GO categories, the counts of domain-GO associations with the highest probability bin (0.9 to

1.0) of MF terms showed the largest decrease when compared with the results with full training data, $\mathcal{D}$ (orange bar). This is probably because MF terms (e.g., enzymatic function) are associated with a domain at a residue level unlike BP and CC, which are more contextual[44].

**Fig. 2 Domain and GO associations using DomainGO-prob. a** Functional consistency of domain embeddings. The domain functional similarity was quantified by the Jaccard Index of GO terms relative to the Manhattan distance of domain embeddings, computed on 100,000 random domain pairs. Three GO categories, MF, BP, CC, are separately shown. **b** Functional coherence in the protein level. 1,000,000 random Protein pairs were split into bins based on their embedding distance and the mean funSim score for each bin was plotted. Bins with <100 proteins were discarded. The last bin includes protein pairs with a distance >25. The size of circles indicates the number of protein pairs in the bin and the color of a data point indicates the standard deviation of the funSim score. **c** Predicted scores of GO terms for domains in 34,832 InterPro2GO entries. The score distribution GO terms for domains were taken from DomainGO-prob. We included the scores from both the standard model (trained on the entire dataset) and the model trained in the adversarial manner (trained after removing the InterPro2GO pair information), which are represented with orange and blue bars, respectively.

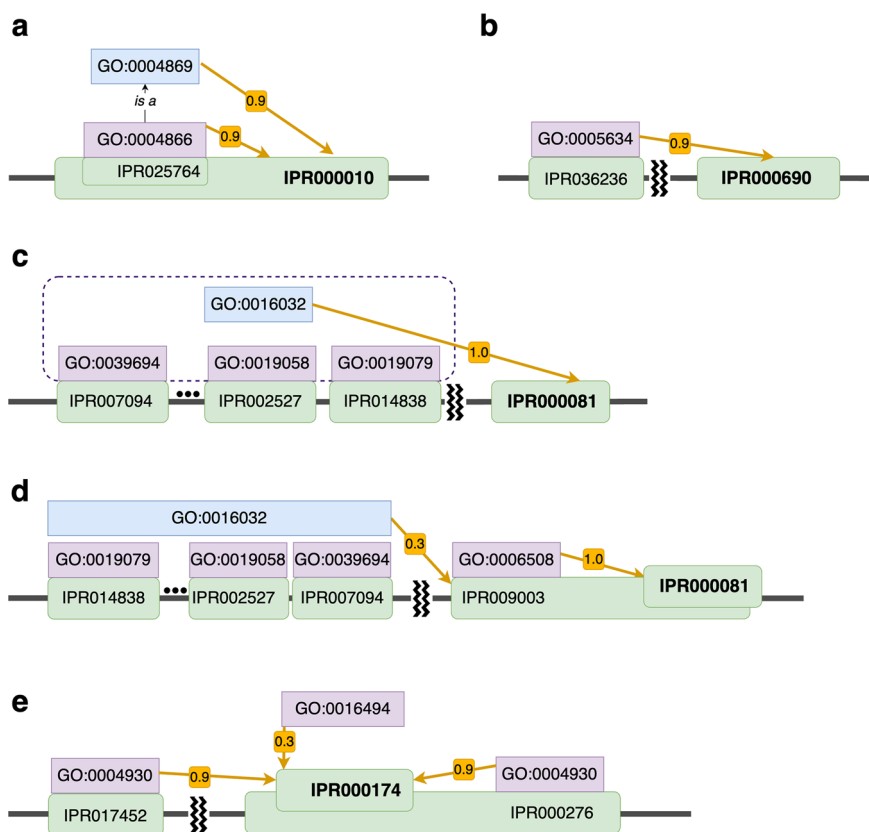

**Fig. 3 Examples to explain how domain-GO associations learned by DomainGO-prob can be interpreted.** The target domain in discussion in each example are shown in bold. Yellow arrows indicate the GO term (blue) that was transferred to the target domain. The associated number with the arrow is the predicted probability that the domain has the GO term. Other GO terms in discussion are colored purple. **a** An example where a GO term was obtained from subdomain. GO:0004866 is a direct parental term of GO:0004869 with "is a" relationship. **b** An example of learning GO terms from co-occurred domains. **c** Examples that a GO term was obtained from multiple co-occurring domains that have a common context. **d** An example where DomainGO-prob was able to distinguish correct and incorrect GO terms that exist in co-occurring domains. **e** An example where the contextual information is not sufficient to retrieve a more specific GO term.

**Examples of domain-GO associations learned by the network.** In this section, we discuss several examples that illustrate how DomainGO-prob learns domain-GO associations. We used the aforementioned adversarial version, i.e., the model trained with $\mathcal{D}'$, and we examined how the model likely learned the GO terms solely from the co-occurrence of different domains. The examples show that DomainGO-prob recovered the correct domain and GO relationships in InterPro2GO from other domain and GO associations in a way that is consistent with the hierarchical and associative relationships of domains and GO terms. This is analogous to the way grammatical structures and word relations aid in masked language modeling[33] in NLP.

The first example (Fig. 3a) is IPR000010, a domain in cysteine protease inhibitors[45], which was annotated with GO:0004869 (cysteine-type endopeptidase inhibitor activity) and GO:0004866 (endopeptidase inhibitor activity) both with a high probability of 0.9 by DomainGO-prob. The GO term GO:0004869 in the MF

category represents binding to and preventing the activity of cysteine-type endopeptidase. Looking at the domain structure, IPR000010 has three subdomains, among which only one subdomain, IPR025764, a Fetuin-B-type cystatin domain, has an annotated GO term with experimental evidence, GO:0004866 (endopeptidase inhibitor activity)[46]. From this domain hierarchy, in addition to GO:0004866, a child term, GO:0004869 with an 'is a' relationship with GO: 0004866, was correctly transferred to IPR000010 by DomainGO-prob.

The second example is a recovered annotation of a CC term, GO:0005634, which represents nuclear localization, with a probability of 0.99 assigned to IPR000690. In InterPro2GO, GO:0005634 is the only CC term associated with this domain. IPR000690 is Matrin/U1-C, C2H2-type zinc finger, which co-occurs with the homologous superfamily IPR036236 in 86.6% of protein sequences (Fig. 3b). IPR036236 is zinc finger C2H2-type superfamily, and 98.6% of its proteins are annotated with the CC

term of nuclear localization. For instance, the protein A5PJN8 has both the two InterPro entries and is also annotated with the GO:0005634 term. Therefore, DomainGO-prob extracted the CC term from the co-occurrence of these domains in proteins and correctly annotated IPR000690.

The next example in Fig. 3c illustrates the transfer of a GO term from multiple co-occurring domains in proteins. DomainGO-prob annotated IPR000081 with the function GO:0016032 (viral process) with a probability of 1.0, which refers to a multi-organism process by a virus. All proteins with this domain (for example, P03303) also have domains IPR007094 (encoded in RNA-containing viruses), IPR001205 (found in RNA viruses), IPR000605 (found in DNA viruses), IPR029053 (forms icosahedral virus shell), IPR002527 (alters membrane permeability), IPR014838 (poliovirus replication) or 8 other domains related to various viral activities. Although not all such co-occurring domains have exactly GO:0016032, they all have related terms, such as GO:0039694 (viral RNA genome replication). Therefore, DomainGO-prob was able to learn the viral process function GO:0016032 by combining such supplementary information.

Some domains are responsible for multiple different functions. For example, the domain IPR000081, which has just been analyzed for viral activity in the previous example, was also correctly assigned with proteolysis (GO:0006508) by DomainGO-prob with a predicted probability of 1.0 (Fig. 3d). However, this information was not learned from the aforementioned co-occurring domains, but rather from the homologous superfamily IPR009003 (Peptidase S1, PA clan), which all proteins with IPR000081 is a part of. For example, the protein P06209 not only contains the domain IPR000081 but also is a member of IPR009003 homologous superfamily. Cysteine peptidase from IPR009003 hydrolyzes a peptide bond using the thiol group[47] and thus has the GO:0006508 function, which was derived to IPR000081. It should be noted that despite IPR009003 completely overlapping with IPR000081, DomainGO-prob did not associate IPR009003 with viral activity (GO: 0016032). For the domain IPR009003 DomainGO-prob predicted a small probability of 0.33 for GO:0016032 (viral process), which was likely induced from the several co-occurring domains involved in viral activities, for example, IPR007094, IPR002527, IPR014838. On the contrary, the actual function for IPR000081, i.e., GO:0006508 was predicted with a probability of 1.0. Therefore, for this example DomainGO-prob was capable of contrasting between complementary information.

There are cases where DomainGO-prob failed to associate GO terms to domains. For instance, in Fig. 3e, IPR000174 represents two different Chemokine receptors from the CXC family, namely CXCR1 and CXCR2[48]. Therefore, proteins from this family are annotated with the function GO:0016494 (CXC chemokine receptor activity). Since Chemokine receptors are part of the G protein-coupled receptor (GPCR) family, proteins from the IPR000174 family (for example, P21109) are also members of IPR000276 (G protein-coupled receptor, rhodopsin-like) and have the IPR017452 (GPCR, rhodopsin-like, 7TM) domain. Although this context provides information about the GPCR family, it is difficult to narrow it down to the CXC family, without individual precise information. This precise information is absent in our adversarial mode of training. As a result, DomainGO-prob predicted a low score of 0.34 for GO:0016494 but managed to assign GO:0004930 (G protein-coupled receptor activity) to IPR000174 with 0.95 probability from the co-occurrence of IPR017452.

**Comparison with large protein language models in GO function prediction.** We evaluated the performance of DomainGO-prob embedding in comparison with 12 large Protein Language Models (PLMs) following the benchmark study performed by Unsal et al.[49]. The 12 PLMs we compare against are ProtT5-XL[30], ProtALBERT[30], SeqVec[50], ProtBERT-BFD[30], ESM-1b[31], ProtXLNet[30], TAPE-BERT-PFAM[51], CPCPProt[52], MSA-Transformer[53], UniRep[54], Learned-Vec[55], and ProtVec[56]. These PLMs were trained on unsupervised tasks such as predicting a segment of masked residues given the rest of the protein[30] or predicting the next residue from all the residues before it[50], on a large protein sequence dataset, e.g. the entire UniProt. Supplementary Table 1 summarizes how these PLMs were trained.

To use a PLM for GO prediction, Unsal et al. converted the residue-level embedding to protein-level by computing the mean of the embeddings along the residues and used a linear Support Vector Machine model. The benchmark by Unsal et al. was performed on the PROBE benchmark dataset they constructed, which provides GO terms of different difficulties to predict. In the Probe dataset, GO terms are divided into three categories based on the frequency in the PROBE benchmark dataset (low, middle, high having 2–30, 100–500 > 1000 annotated proteins, respectively) and specificity (shallow, normal, specific for the ontologies being within the depth of $1/3^{rd}$, $2/3^{rd}$ and bottom rest, respectively). Therefore, based on frequency and specificity, $3 \times 3 = 9$ groups of GO terms can be constructed for the three GO categories, i.e. $3 \times 9 = 27$ groups. Among them, as there were no GO terms that fall under the high-specific group, the benchmark ended up with 25 groups. For each group, at most 5 GO terms were selected based on dissimilarity according to the Lin's similarity measure[42], which resulted in a total of 117 GO terms to predict. The PROBE dataset contains 19,995 human proteins clustered at a 50% identity cutoff and only experimental GO annotation. The human proteins falling under these criteria were used for benchmarking GO function prediction by under-going a 5-fold cross-validation test. Unsal et al. provided a convenient CodeOcean distribution (https://PROBE.kansil.org, version November 3, 2022), where given the embeddings of the test proteins, GO predictions are made, and the performance is evaluated on the PROBE dataset. We used it to test our DomainGO-pair-based protein embedding (Eq. 4).

For this benchmark, we trained the domain and GO embeddings (Eq. 2) for the three GO categories separately on Swiss-Prot, after removing all the human proteins. We removed these proteins to avoid overlap between the test proteins and the proteins used for training. However, note that the PLMs we compared against almost certainly have these human proteins in their training set, as they used an entire public protein sequence dataset for training. Since our embedding dimension is only 256, which is quite small compared to the PLMs we compared against, we concatenated the embeddings from the three GO categories and performed mean normalization to balance them. This resulted in a 768-dimensional protein embedding vector, as follows:

$$Embedding(p) = MeanNorm(Concat(D_{MF}(p), D_{BP}(p), D_{CC}(p)) \tag{7}$$

Here, $D_{MF}(p), D_{BP}(p), D_{CC}(p)$ are computed embeddings for a protein $p$ for MF, BP, CC sub-ontologies, respectively (using Eq. 4.)

The results are presented in Fig. 4a, where we compared the GO prediction performance of our model (Eq. 7) with 12 models on the PROBE benchmark. The numerical values are provided in Supplementary Table 2. Our model based on DomainGO-prob outperformed all the PLMs in all three categories. For MF, BP, and CC, DomainGO-prob resulted in 0.02, 0.06, and 0.06 higher weighted F1 scores than ProtT5-XL, the previous top method, respectively, and 0.04 when the average across the three GO

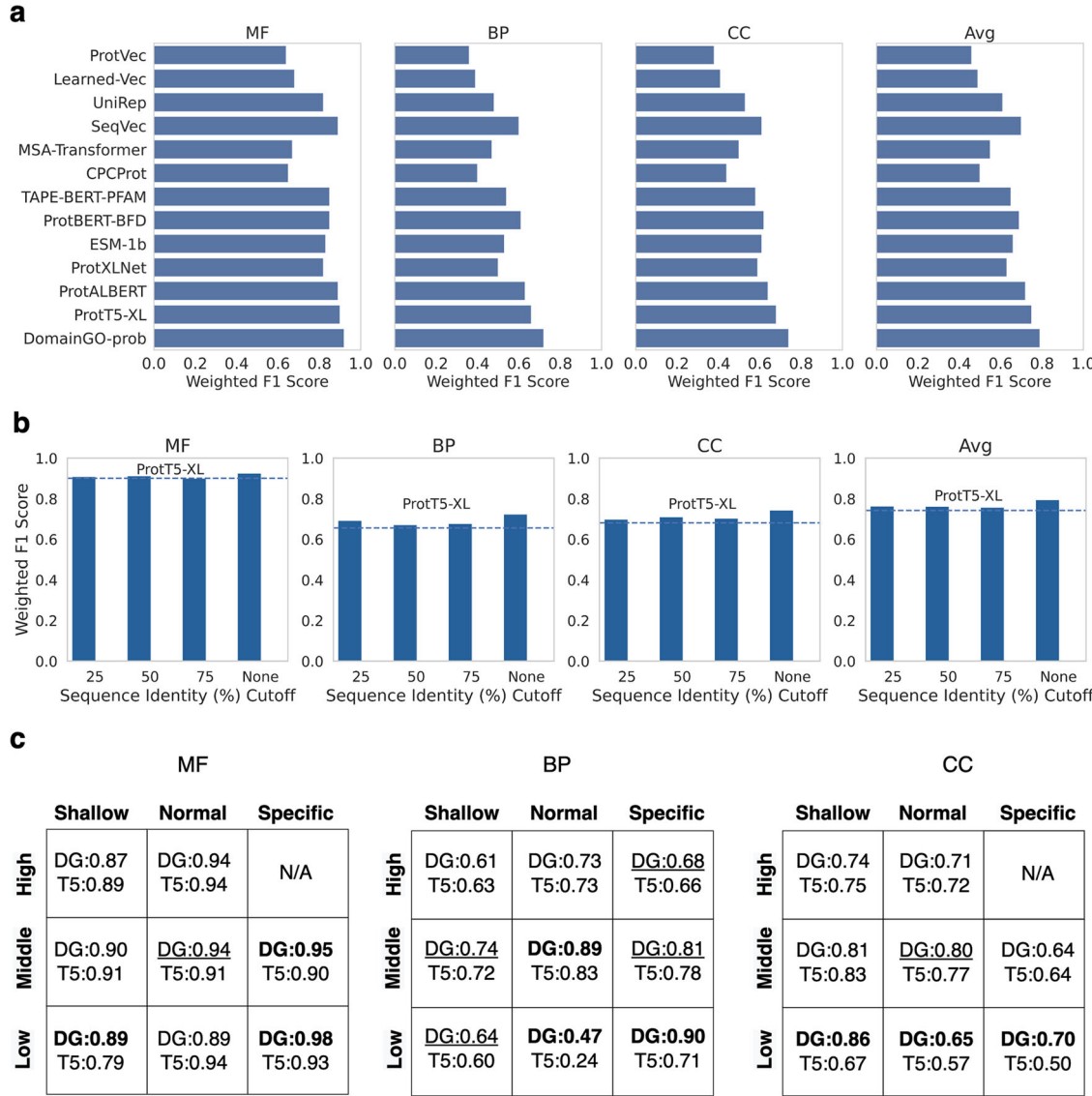

**Fig. 4 Comparison with 12 large PLMs in the PROBE benchmark. a** The weighted F1 Score obtained by our model that is based on DomainGO-prob with 12 PLMs on the PROBE benchmark. To compute the weighted F1 score, the values of F1 score for the individual GO terms were averaged, weighted by the number of samples having that particular GO term, for all three GO categories and the average across the categories. **b** Analysis of the performance of DomainGO-prob when trained on non-redundant datasets. The model with DomainGO-prob was retrained on training datasets after removing proteins with 75%, 50%, and 25% sequence identity to the test proteins. None indicates the result using the original training set without further removing training proteins. The dashed line shows the weighted F1 score of ProtT5-XL. **c** Comparative performance on the nine groups of GO terms with different difficulty levels against the best-performing ProtT5-XL. DG, prediction with the model (Eq. 7) using DomainGO-prob. N/A indicates that there is no GO term in that particular group. DG is shown in bold and underlined when the improvement over ProtT5-XL is greater than and less than 5%, respectively.

categories was considered. Notably, this improvement was obtained from a much simpler model with 768-dimensional embeddings with merely a fraction of the parameters of the PLM models by adopting a functionally informed learning protocol. As shown in Supplementary Table 1, ProtT5-XL has 1024-dimensional embeddings and was trained with a network with 3 billion parameters, while the three networks we used (Eq. 7) have only 31 million parameters in total.

An important consideration when training machine learning models for protein analysis is to remove redundancy, i.e., similar sequences from the training set relative to the test set. Therefore, although we have already omitted human protein sequences from our training dataset, we retrained our models after removing proteins with 75%, 50%, and 25% sequence identity with the test set using MMseqs2[57]. The results are shown in Fig. 4b, in

comparison with ProtT5-XL. As expected, the F1 score decreased slightly as more sequences were removed. However, this is most likely due to the fact that we were losing some domain and GO association information when we removed the proteins. Nevertheless, in all cases, our embedding performed better than that of ProtT5-XL, even at the identity cutoff of 25%.

In Fig. 4c, we examined how performance changed when considering GO terms of varying levels of difficulty to predict. Weighted F1 scores for the different GO groups in the PROBE benchmark, classified by GO depth and frequency, were separately shown. As we moved from high to low frequency or shallow to specific GO terms, the classification task became more difficult. We compared our model's performance with the best-performing PLM, ProtT5-XL. Even in this evaluation, it was evident that our model substantially outperformed ProtT5-XL.

**Table 1 Comparison with state-of-the-art function prediction methods.**

| Method | Features used | $F_{max}$ | | | AUPR | | | $S_{min}$ | | |
|---|---|---|---|---|---|---|---|---|---|---|
| | | MF | BP | CC | MF | BP | CC | MF | BP | CC |
| BLAST | BLAST | 0.627 | 0.407 | 0.625 | 0.427 | 0.272 | 0.412 | 5.503 | 25.918 | 9.351 |
| DeepGOCNN | Sequence | 0.589 | 0.337 | 0.624 | 0.565 | 0.271 | 0.623 | 6.417 | 27.235 | 10.617 |
| DeepGOPlus | Sequence, BLAST | 0.661 | 0.419 | 0.655 | 0.667 | 0.342 | 0.663 | 5.407 | 25.603 | 9.374 |
| DeepGOZero | Domain | 0.662 | 0.396 | 0.662 | 0.668 | 0.337 | 0.645 | 5.322 | 25.838 | 9.834 |
| DeepGOZero + BLAST | Domain, BLAST | 0.655 | 0.432 | 0.675 | 0.665 | 0.356 | 0.654 | 5.337 | 25.439 | 9.391 |
| DeepGraphGO | Domain, PPI | 0.671 | 0.418 | *0.679* | 0.647 | 0.364 | 0.669 | 5.374 | 25.866 | 9.165 |
| TALE+ | Sequence, BLAST | 0.466 | 0.382 | 0.661 | 0.441 | 0.31 | 0.681 | 8.136 | 26.308 | 9.599 |
| NetGO2.0 (Server) | Domain, Kmer, RNN, PPI, BLAST, Pubmed, Frequency | **0.698** | 0.431 | 0.662 | 0.701 | 0.343 | 0.627 | 5.187 | *25.076* | 9.473 |
| Domain-PFP | Domain | *0.675* | 0.41 | 0.675 | 0.676 | 0.344 | 0.697 | 5.259 | 25.838 | 9.709 |
| DPFP + BLAST | Domain, BLAST | 0.674 | *0.434* | 0.681 | *0.693* | *0.367* | **0.717** | *5.188* | 25.11 | *9.239* |
| DPFP + PPI | Domain, PPI | 0.666 | 0.435 | 0.673 | 0.689 | 0.379 | 0.675 | 5.404 | 25.002 | 9.35 |
| DPFP + BLAST + PPI | Domain, BLAST, PPI | 0.685 | **0.452** | **0.686** | **0.718** | **0.393** | *0.687* | **5.146** | **24.292** | **9.084** |

The best, 2nd best, and 3rd best results are indicated by bold, underline, and italic, respectively. We also include the features used by the predictors. *DPFP* Domain-PFP, *Kmer* the K-mer distribution in the protein sequences, *RNN* protein sequence embedding computed by a recurrent neural network, *Frequency* the frequency of the GO terms in the database.

Interestingly, the margin of the advantage of our model increased as we considered more difficult GO groups. In most of the easier cases, DomainGO-prob was at least similar to or slightly better than ProtT5-XL. In difficult cases, a substantial improvement was observed. For example, for low-frequency and specific CC terms, DomainGO-prob was 20% better. It is apparent that although the PLM was able to comprehend frequent GO terms from unsupervised learning on a large volume of protein sequence data, such models failed to account for rare GO terms and suffered from limited specificity. On the contrary, our self-supervised learning approach seemed to decipher the functional identity of proteins better, regardless of the rarity and specificity of the GO terms to a degree.

**GO function prediction by Domain-PFP in comparison with existing methods**. Subsequently, we benchmarked the GO prediction performance of Domain-PFP (Eq. 5) on the NetGO dataset[16] to compare it with state-of-the-art protein function prediction methods from recent literature. We used the data split of the NetGO dataset into training, validation, and test sets provided in the work of DeepGOZero[22], who followed the same data split protocol of NetGO2.0[16]. The NetGO dataset consists of 64,279, 91,443, and 83,004 proteins for MF, BP, and CC categories, respectively, with a specified training, validation, and test set split (Supplementary Table 3). We trained DomainGO-prob on the NetGO training dataset and created a weighted K Nearest Neighbor (KNN) model based on the learned embedding. The number of K-neighbors used for MF, BP, and CC was 1000, 800, and 1200, respectively, which were tuned based on the performance on the validation data split of the NetGO benchmark (Supplementary Fig. 1). The performance of the various models on the test dataset is presented in Table 1. The evaluation results of the existing methods, from BLAST to NetGO2.0 (Server) in Table 1, were taken from the paper of DeepGOZero[22].

DeepGOPlus[18] infers protein functions through a combination of DiamondBLAST[58] and DeepGOCNN, which employs a 1D convolutional neural network to predict GO from the amino acid sequence. TALE+[19] similarly fuses DiamondBLAST with sequence representation learned from a Transformer. Other top-performing methods are either based on domain information or used as a component. For instance, DeepGOZero[22] leverages a model-theoretic approach to predict ontologies from InterPro domains, which can be further improved by incorporating DiamondBLAST. DeepGraphGO[21] associates InterPro features with protein-protein interaction (PPI) networks employing a graph convolutional neural network. NetGO2.0[16] is an all-encompassing ensemble method that incorporates BLAST, domain, PPI, GO term frequency, PubMed publications, and sequence information both in form of k-mers and embedding. Among the existing methods, NetGO2.0 has shown the highest evaluation values for MF and the best $S_{min}$ value in BP[36] (note that the NetGO2.0 results are from the current server, ran by the authors of DeepGOZero in their paper).

In the latter half of Table 1 we show results by Domain-PFP and Domain-PFP that incorporate BLAST and PPI information to compare with the other state-of-the-art methods that combine diverse information sources. The scores of GO term *j* for protein *i* from BLAST and PPI information are defined as

$$S_B(p_i, GO_j) = \frac{\sum_{p_k \in D} I(p_k, GO_j) \times B(p_i, p_k)}{\sum_{p_k \in D} B(p_i, p_k)} \quad (8)$$

$$S_N(p_i, GO_j) = \frac{\sum_{p_k \in D} I(p_k, GO_j) \times \omega(p_i, p_k)}{\sum_{p_k \in D} \omega(p_i, p_k)} \quad (9)$$

Here, $B(p_i, p_k)$ and $\omega(p_i, p_k)$ are the bit-score from Diamond-BLAST with 'more-sensitive' setting[58], and edge weight from STRING PPI network (ver. 11.0)[59], respectively. We used the same STRING version as DeepGraphGO[21].

The final score is a simple average of the terms from the three sources:

$$S(p_i, GO_j) = \frac{S_D(p_i, GO_j) + I_B(p_i) S_B(p_i, GO_j) + I_N(p_i) S_N(p_i, GO_j)}{1 + I_B(p_i) + I_N(p_i)}$$

$$(10)$$

$I_B$ and $I_N$ are identity functions, which results in 1 if BLAST and String Network matches are found for the protein *i*, respectively.

We compared the performance of the methods using the three CAFA evaluation metrics, namely $F_{max}$, AUPR, and $S_{min}$[36] (see Methods). $F_{max}$ computes the maximum possible protein-centric F1 score, overall prediction thresholds. AUPR, the area under PR curve, on the other hand, is a suitable metric for imbalanced data and penalizes the false positive predictions, which is highly applicable to function prediction[16]. Finally, $S_{min}$ is a measure of semantic distance between predicted and actual annotation, based on the information content of the individual GO terms[18], i.e., this metric indicates the capability of predicting rare GO terms.

Firstly, we compared Domain-PFP with sequence-only or domain-based methods, e.g., DeepGOCNN and DeepGOZero. This is a fair comparison as the base Domain-PFP uses only domain information which is inferred from sequence information. It can be observed from the table that Domain-PFP outperforms these methods in terms of $F_{max}$, AUPR, and $S_{min}$ in all the three sub-ontologies. Notably, Domain-PFP achieved an AUPR of 0.697 for CC, whereas DeepGOZero, a recent method based on domain information scored 0.645, i.e., a large improvement of 0.052. In terms of $F_{max}$, Domain-PFP outperformed DeepGOZero and DeepGOCNN by achieving 0.013–0.014 and 0.051–0.086 higher scores, respectively.

Adding different features generally improves the performance of function prediction. When BLAST information was combined, Domain-PFP improved the overall performance, except for a slight drop of 0.001 in $F_{max}$ for MF. $F_{max}$ for BP increased from 0.41 to 0.434, and AUPR for CC increased from 0.697 to 0.717. Furthermore, Domain-PFP with BLAST consistently outperformed DeepGOZero+BLAST, which also uses the same information, in all 9 metrics. For example, DeepGOZero+BLAST achieved AUPR scores of 0.665, 0.356, and 0.654 for MF, BP, and CC, respectively, whereas Domain-PFP + BLAST achieved 0.693, 0.367, and 0.717, representing improvements of 0.028, 0.011, and 0.063, respectively. When compared with DeepGOPlus or TALE +, both of which use BLAST, the improvements made by Domain-PFP + BLAST appeared consistent as well.

Next, we experimented with including PPI information with Domain-PFP. However, this only improved the performance in BP, as expected, since BP involves multiple related and interacting functions that can be captured by PPIs. On the other hand, the performance of MF and CC was negatively affected. This situation is similar to the findings of NetGO2.0[16], where the authors reported that PPI information performed better than domain information for predicting BP terms but not for MF and CC terms. For example, the $F_{max}$ of MF and CC dropped by 0.009 and 0.002, respectively. Despite this, Domain-PFP + PPI still outperformed DeepGraphGO, a method using domain and PPI information in a much more sophisticated graph neural network, in 5 out of 9 metrics.

Finally, we experimented with integrating both BLAST and PPI simultaneously. This brought improvements in all the metrics except for AUPR of CC. Notably, $F_{max}$ and AUPR of BP improved by 0.042 and 0.049, respectively. This integration of BLAST and PPI features enabled Domain-PFP to consistently perform superior to all the existing methods. For example, the current state-of-the-art method NetGO2.0 was surpassed by Domain-PFP in 8 out of 9 metrics (except for $F_{max}$ for MF). In terms of $F_{max}$ for BP and CC, Domain-PFP achieved 0.021 and 0.024 higher scores, respectively. For AUPR, the improvements were 0.017, 0.050, and 0.060 for MF, BP, and CC, respectively. Similarly, in terms of Smin, Domain-PFP + BLAST + PPI achieved 0.041, 0.784, and 0.389 smaller scores for MF, BP, and CC, respectively, implying that non-trivial GO terms were captured better.

This comparative evaluation with state-of-the-art function prediction methods further supports our self-supervised approach of learning functionally informed representations for protein domains. We observed that a simple KNN model with DomainGO-prob embedding not only outperformed more sophisticated deep learning models (e.g., DeepGraphGO) but also methods with access to more information sources (e.g., NetGO2.0). The only case where we fell behind the previous state-of-the-art, NetGO2.0, is in $F_{max}$ for MF. which we hypothesize is due to the inclusion of Pubmed publication information that is likely to contain precise information vital for MF prediction.

In our evaluation, we have utilized the benchmark compiled by the authors of DeepGOZero, which was derived from the NetGO benchmark dataset, following protocols similar to CAFA. Using this benchmark allowed us to compare our performance against the other recent methods, that were evaluated on this benchmark by the authors of DeepGOZero. We also evaluated Domain-PFP on the original NetGO benchmark. The results of those experiments are presented in Supplementary Table 4. Domain-PFP with BLAST and PPI showed the highest values for all the metrics except for AUPR for CC. For AUPR for CC, Domain-PFP with BLAST showed the highest value. Domain-PFP alone showed a higher score than all the existing methods compared except for $F_{max}$ of MF, where NetGO had the highest score. Compared to $F_{max}$, the improvement in AUPR is more prominent, which can also be observed in the results presented in Table 1.

To assess the performance of Domain-PFP against structure-based protein function predictors we considered two recent methods DeepFRI[8] and GAT-GO[60]. These methods use 3D protein structure information in a graph neural network and protein sequence information with a language model. Both methods were evaluated on a common benchmark dataset, composed of 29,902, 3,323, and 3,416 proteins for training, validation, and testing, respectively. The train and test proteins possess a total of 2,752 GO terms and they are clustered with 40% sequence identity.

We retrained Domain-PFP on this dataset and observed the performance across the three sub-ontologies. The results are presented in Supplementary Table 5. It can be observed that Domain-PFP outperforms much complex graph neural network-based function predictors with access to structural information on all the metrics except for $F_{max}$ in CC and AUPR in MF. The performance of Domain-PFP was further improved by including BLAST predictions, which results in the best score for all the metrics.

**Evaluation on CAFA3 benchmark**. We further evaluated Domain-PFP on the CAFA3 benchmark[36]. We trained the network model of Domain-PFP using the CAFA3 training dataset and evaluated the results using the official evaluation code. The training dataset comprised 66,841 protein sequences annotated before September 2016, with 677, 3992, and 551 MF, BP, and CC GO terms, respectively (Supplementary Table 6). The test set contained 3328 proteins annotated between September 2016 to February 2017. To include sequence similarity information using BLAST in our pipeline, we constructed a new BLAST database with the CAFA3 training sequences. However, we could not use PPI information from the STRING database for this benchmark because STRING v10.a (the version during the competition timeline) lacked sufficient interaction data of the CAFA3 test proteins. We did not perform any additional hyperparameter tuning and kept the same hyperparameters computed from the NetGO benchmark validation data.

The results of Domain-PFP on the CAFA3 benchmark are presented in Fig. 5 in comparison with the top 10 performing methods as published by the organizers of CAFA3[36]. DomainPFP +BLAST consistently showed a higher $F_{max}$ than Domain-PFP alone. For both BP and CC, Domain-PFP + BLAST outperformed the existing methods. Domain-PFP + BLAST achieved a $F_{max}$ score of 0.63 for CC, which is 0.02 higher than the CAFA3 top model, Zhu Lab. For BP, Domain-PFP + BLAST showed a slightly higher $F_{max}$ of 0.398 than the CAFA3 top model ($F_{max}$: 0.397). For MF, our $F_{max}$ score, 0.59, was second to Zhu Lab ($F_{max}$: 0.62), with a substantial margin to the next method, orengo-funfams ($F_{max}$: 0.54). The top method by Zhu Lab

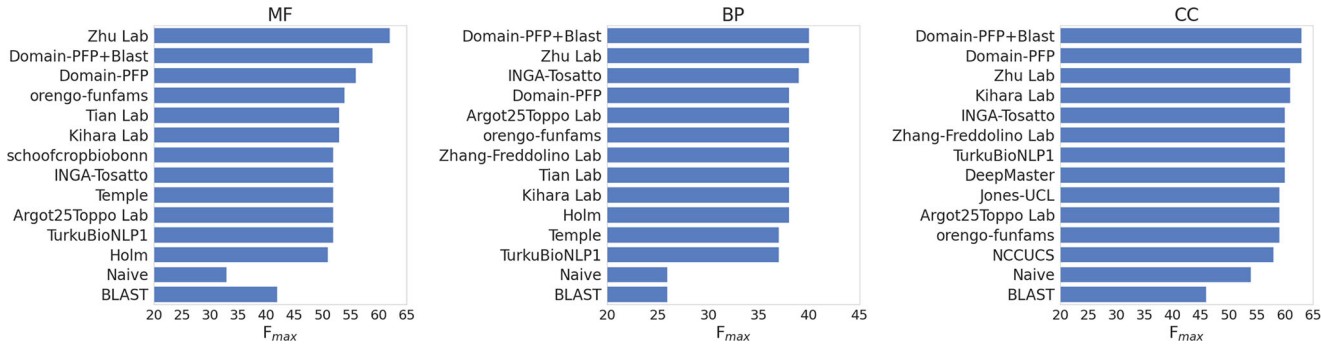

**Fig. 5 Comparison with CAFA3 methods.** The $F_{max}$ for the top 10 methods and the 2 baseline methods (Naïve, BLAST) are presented. All the scores were collected from the official CAFA3 result (Zhou et al.[36]). The $F_{max}$ score of Domain-PFP was also computed using the official CAFA3 evaluation codes.

combined more diverse information using an ensemble approach, including sequence, domain, homology, biophysical information, which likely gave that a competitive edge, similar to NetoGO2.0. We also mention that both DomainPFP and DomainPFP +BLAST showed higher $F_{max}$ scores than DeepGOPlus, which reported $F_{max}$ scores of 0.557, 0.390, and 0.614 for MF, BP, and CC, respectively, in their paper[18].

## Discussion

Despite protein domains carrying the functional signatures of proteins, they have not been used to their full potential to date. Look-up-table-based domain to GO assignments tend to lack coverage. On the other hand, deep learning-based approaches using domains as high-dimensional input suffer from limitations in training data and information loss in network bottlenecks. Therefore, our motivation has been twofold: improving coverage and reducing information loss. Based on recent advancements in self-supervised learning, it has become motivating to apply such concepts in protein domain learning to alleviate these issues. Our method follows one of the core concepts of self-supervised learning where pseudo-labels are first learned to initialize model parameters, which are then used to perform the actual task using a supervised or unsupervised method[61]. Our approach is consistent with this definition as we first use the domain-go association probabilities as pseudo labels, which initializes our domain embedding parameters; then, we use this embedding later in a supervised learning protocol and we predict the functions of the proteins. This strategy also holds in the benchmarks on the NetGO and CAFA3 dataset we performed. To the best of our knowledge, this work is the first to apply self-supervised learning in the domain of protein function prediction. Based on co-occurrence contextual information between domain and GO terms, we devise embeddings for domains so that functionally related domains have similar embeddings. Since co-occurrences were learned from entire protein sequences, the domain embedding, DomainGO-prob, encodes GO associations that are not explicitly described in the domain database. Remarkably, our rather simple model, Domain-PFP, along with BLAST and PPI information, demonstrated superior performance over all state-of-the-art function predictors.

One likely limitation of this work could be the case of unknown domains. All existing methods based on protein domains fail to predict anything if the domain seen during inference was absent in the training data, in which case they predict a default value. This limitation can possibly be resolved by generating the functionally aware domain representation and localization end-to-end from the protein sequence directly using a larger deep learning model. Another limitation is that the current protein embedding considers domains in a protein equally (Eq. 4), although each domain may have different levels of contribution to protein

function. Also, the order of appearance of domains in a protein is not considered, which is known to be relevant to function[13]. To address these, attention mechanism maybe applicable. These are improvements we wish to explore in our subsequent works. In the current work, we practically alleviated these issues by augmenting with BLAST and PPI information-based predictions.

Another possible future direction could be to combine with general protein language models[31], which were shown to perform well in protein tertiary structure prediction and other tasks. Additionally, we wish to analyze the suitability of our model in a zero-shot learning scenario. Specifically, our goal is to train DomainGO-prob on pretrained GO embeddings based on GO tree hierarchy and observe if GO terms absent in the training data can be retrieved this way.

## Methods

**Neural network architecture**. We have designed a neural network to learn the domain-GO co-occurrence conditional probability distribution (Fig. 1a). The domain and GO terms are received as one-hot-encoded inputs, which are passed through two separate embedding layers to generate the 256-dimensional domain and GO embeddings, respectively. Then, from the computed domain and GO embeddings, we calculate the Hadamard product as a measure of correlation between the two types of embeddings and pass them through a densely connected layer of 128 neurons. The neurons are regularized through dropout ($p = 0.05$) and activated by RELU. Finally, we use a linear layer to predict the $p(domain|GO)$ score. The domain embedding matrix is extracted to generate the representations of domains. In order to increase the sparsity of the domain embeddings, we apply L1-regularization on that embedding layer ($\lambda = 0.1$).

**Network training**. Similar to word2vec embedding training[62], we have a comparatively much larger number of domain-GO terms out of context, i.e., $p(GO|domain) = 0$. Thus, we employed negative sampling by randomly selecting 1000–2000 non co-occurring GO terms for each domain. The network was trained by minimizing the MSE (mean squared error) loss with Adam optimizer[39] with a learning rate of 0.001 (the other parameters were kept as default) and a batch size of 163,840 for 200 epochs. 20% of domain-GO pairs, which were randomly selected, were used as the validation set. The experiments were performed 10 times and the best model based on validation performance was selected.

**funSim score**. *funSim* score is popularly used for quantifying similarity of GO term annotation of two proteins[29,63]. *funSim* score uses the relevance semantic similarity score $sim_{REL}$ for the

similarity of GO terms of the same category[42]:

$$sim_{Rel}(GO_1, GO_2) = \max_{go \in Ancestors(GO_1, GO_2)}$$
$$\left( \frac{2 \log p(go)}{\log p(GO_1) + \log p(GO_2)} \times (1 - p(go)) \right) \quad (11)$$

where common ancestral GO terms of $GO_1$ and $GO_2$ are explored to maximize the score and p(GO) is the probability of GO term in the entire Swiss-Prot database. Then, a set of GO annotations in a GO category for two proteins, $a$ and $b$, are defined as

$$GO_{score}(protein_a, protein_b) = \max\left( \frac{1}{N} \sum_{i=1}^{N} \max_{1 \le j \le M} s_{ij}, \frac{1}{M} \sum_{j=1}^{M} \max_{1 \le i \le N} s_{ij} \right) \quad (12)$$

where $s_{ij}$ is $sim_{REL}$ score of $GO_i$ and $GO_j$ of $protein_a$ and $protein_b$, respectively, computed in an all-vs-all fashion.

Finally, *funSim* score is the average of the $GO_{score}$ from the three GO categories.

$$funSim = \frac{1}{3}\left[ \left( \frac{MF_{score}}{\max(MF_{score})} \right)^2 + \left( \frac{BP_{score}}{\max(BP_{score})} \right)^2 + \left( \frac{CC_{score}}{\max(CC_{score})} \right)^2 \right] \quad (13)$$

*funSim* score ranges from 0 to 1 with 1 as the maximum score.

**Evaluation metrics.** For the PROBE benchmark, similar to the original benchmark by Unsal et al.[49], we used Weighted F1 Score as the evaluation metric. The values were computed using their official CodeOcean distribution.

To compare with state-of-the-art methods, we used the CAFA protein-centric evaluation metrics F$_{max}$, Smin, and AUPR[2]. We used the same evaluation codes as used by[22] to ensure consistency.

F$_{max}$ is the maximum possible protein-centric F1 score, computed over all prediction thresholds.

$$F_{max} = \max_{0 \le \tau \le 1} \frac{2 pr(\tau)\, re(\tau)}{pr(\tau) + re(\tau)} \quad (14)$$

Here, $pr(\tau)$ and $re(\tau)$ Are precision and recall scores, respectively, computed at the cut-off value of $\tau$. The precision and recall values are computed as

$$pr(\tau) = \frac{1}{h(\tau)} \sum_{j=1}^{h(\tau)} \frac{\sum_i I\left(S(G_i, P_j) \ge \tau\right) . I(G_i, P_j)}{\sum_i I\left(S(G_i, P_j) \ge \tau\right)} \quad (15)$$

$$rc(\tau) = \frac{1}{N_T} \sum_{j=1}^{N_T} \frac{\sum_i I\left(S(G_i, P_j) \ge \tau\right) . I(G_i, P_j)}{\sum_i I\left(G_i, P_j\right)} \quad (16)$$

Here, $N_T$ is the total number of proteins and $h(\tau)$ is the number of proteins with a prediction score no smaller than $\tau$ for at least one GO term. $I$ is the identity function which returns 1 if the condition is true, 0 otherwise. $I(G_i, P_j)$ therefore, implies if the protein $P_j$ whether has the GO term $G_i$ or not. $S(G_i, P_j)$ denotes the prediction score of $P_j$ having the $G_i$ term.

The area under precision-recall curve, i.e., AUPR score is computed from the computed precision and recall scores using the trapezoidal rule.

$$AUPR = \frac{\Delta x}{2}\left( f(x_0) + 2f(x_1) + 2f(x_2) + \dots + 2f(x_{N-1}) + f(x_N) \right) \quad (17)$$

Here, $x_0, x_1, \dots, x_N$ are various recall values, whereas $f(x_0), f(x_1), \dots, (fx_N)$ are values of precision at those recalls, and $\Delta x$ is the step size.

$S_{min}$ is a measure of semantic distance between the ground truth and prediction annotations based on information content of the GO classes. The information content IC(c) for a class c is computed based on the annotation probability of class c relative to its parent class $P(c)$

$$IC(c) = -\log(probability(c|P(c))) \quad (18)$$

The two terms remaining uncertainty ($ru$) and average misinformation ($mi$) are defined as

$$ru(t) = \frac{1}{n} \sum_{i=1}^{n} \sum_{c \in \{T_i - P_i(t)\}} IC(c) \quad (19)$$

$$mi(t) = \frac{1}{n} \sum_{i=1}^{n} \sum_{c \in \{P_i(t) - T_i\}} IC(c) \quad (20)$$

The value of $S_{min}$ is computed as

$$S_{min} = \min_t\left( \sqrt{ru(t)^2 + mi(t)^2} \right) \quad (21)$$

## Data availability

The embeddings of the proteins from the PROBE benchmark dataset, computed by DomainGO-prob, GO term prediction by DomainPFP on the CAFA3 dataset, trained DomainGO-prob model weights and Domain-PFP KNN models are accessible at https://github.com/kiharalab/Domain-PFP and can also be downloaded from https://kiharalab.org/domainpfp/ and figshare[64]. All other data can be obtained from the corresponding author upon reasonable request.

## Code availability

The Domain-PFP program is freely available for academic use from GitHub at (https://github.com/kiharalab/Domain-PFP). The snapshot of the code at the time of the publication is also made available at Zenodo[65]. Furthermore, the program is available to run on Google Colab Notebook (bit.ly/domain-pfp-colab).

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

## Acknowledgements

This work was partly supported by the National Science Foundation (DBI2003635, DBI2146026, IIS2211598, DMS2151678, CMMI1825941, and MCB1925643) and by the National Institutes of Health (R01GM133840, 3R01 GM133840-02S1).

## Author contributions

D.K. conceived the study. N.I. designed and implemented the domain-based protein embedding and Domain-PFP. N.I. performed the computation. N.I. and D.K. analyzed the data. Y.K. participated in function evaluation on the CAFA3 dataset. N.I. drafted the manuscript and D.K. edited it. All authors read and approved the manuscript.

## Competing interests

The authors declare no competing interests.
