## [Peer review file · Communications Biology]

Reviewers' comments:

Reviewer #1 (Remarks to the Author):

This paper presents Domain-PFP, a protein function prediction method using function-aware domain embeddings. The authors employ a self-supervised protocol to derive functionally consistent representations of domains by learning domain-Gene Ontology (GO) cooccurrences and associations. Specifically, the authors devise embeddings for domains so that functionally related domains have similar embeddings. The author carried out various experiments and discussed associations between Domain and GO and how Domain-GO associations learned by DomainGO-prob can be interpreted. And evaluations showed that protein representations using domain embeddings are superior to those of large-scale protein language models in GO prediction tasks. The authors also discuss the limitations of protein domain approaches: all existing protein domain-based methods fail to predict anything if the domain seen during inference does not exist in the training data. The authors suggest exploring larger deep-learning models to generate feature-aware domain representations. The following are some comments to improve the paper.

Major

1. NetGO 2.0 dataset used in this work is different from the one provided in the original paper of NetGO 2.0, which gives the url of google driver. The original data set should be used for evaluation.
2. As mentioned in section "Comparison with large protein language models in GO function prediction", a protein-level embedding contains 3 embeddings from MFO, BPO, and CCO, respectively. Why are there three representations of each protein? Are the three embeddings per domain trained from MFO, BPO, and CCO, respectively?
3. Different protein domains do not contribute equally to the final function of the protein. Compared with using the average method to integrate the embedding of the protein domain into the embedding of the protein, is it better to add weights?
4. For NetGO dataset and CAFA3 datasets, is the protein embeddings' dimension 256 or 768 when predicting a GO term?

Minor

1. There are some format issues, such as most equations are right-aligned, but some are centered.

2. In eq 6, 'GO Terms A' creates ambiguity and tends to make the readers think that it is a set related to GO Term A rather than a set related to domain A. It is suggested that the authors revise it.

3. Figure B in Figure 2 is difficult to understand. It is more appropriate to use the size of the circle to represent "Number of protein pairs", and use different colors to represent the "Std.

funSim Score".

4. The presentation form of Figure 5 is not very good. It is difficult to see the specific numerical differences at a glance, and such an X-axis is very inconvenient to read. Perhaps

you can swap the X and Y axes (Just like Figure 4. a). And the range of F-max on the axis can start not from 0, but higher (say 20).

Reviewer #2 (Remarks to the Author):

Comments to the authors

Because of the rapid increase in the number of proteins and the diversity of their functions, developing methods for the automatic and accurate prediction of protein functions has become an important issue. For this issue, the authors proposed their new function prediction method, Domain-PFP, which uses function-aware domain embedding representations of proteins, based on domain-GO association learned in a self-supervised manner. After they learned pseudo-labels of GO prediction probability from individual domain terms, they derived a representation of domains consistent with functional information to characterize protein sequences and used that representation to predict the protein functions. They found that such representations were functionally meaningful because their embedding distance showed correlation with functional similarity of GO terms, and showed that protein representations using their domain embeddings are superior to those of large-scale protein language models (PLMs) in GO prediction tasks, despite having only a fraction of the PLM complexity. Finally, they used a KNN classifier, with their learned embeddings, along with sequence similarity and PPI information, and demonstrated that their method exhibited competitive performance when compared with top-scoring methods in the CAFA3 evaluation. According to results reported by the authors, Domain-PFP seems effective, though their model has only a fraction of the PLM complexity.

Nowadays, structure prediction methods such as AlphaFold enable us to obtain more or less accurate structure models for most proteins, meaning that we can use them for structural information for large-scale protein function annotations. In fact, methods such as DeepFRI and GAT-GO have been developed. Consequently, including a systematic comparison with such methods is expected to be helpful for readers.

In the manuscript, the authors stated that “We used self-supervised learning because it can directly learn domain and GO co-occurrence from abundant protein sequences and thus is able to alleviate the low-coverage problems of domain information”, but described that “We intended to keep the function f simple so that the domain embeddings could effectively learn functional relevance, rather than letting the function f learn the correlation between domain and GO term co-occurrence”. It seems that these might not be mutually consistent. We presume that the authors will discuss these points more carefully.

Reviewer #3 (Remarks to the Author):

This manuscript introduces a domain-based method to predict protein function (GO terms), which differs from the previous domain-based methods mainly in using a pre-training step to learn the domain embeddings by modeling the domain-GO cooccurrences and associations. The overall framework is interesting, and the paper is well written. However, some details in the experiments are not very clear, and the discussions of the latest methods should be updated to be more comprehensive. Below are some specific comments that the authors may find useful.

1. When comparing Domain-PFP with other methods in the NetGO 2.0 and CAFA3 datasets, did the authors use the training sets to pre-train DomainGO-prob? Please add more details explicitly. If not, did the authors use all annotations in Swiss-Prot for pre-training? This will lead to the data leakage problem. If yes, I don't think using the GO annotations from the training sets to learn GO-domain associations can be claimed as “self-supervised”. I suggest the authors rephrase some of the sentences.
2. Please update the introduction or comparison to include more latest methods, e.g., language-model-based function predictor SPROF-GO [doi 10.1093/bib/bbad117].
3. As shown in Table 1 and Figure 5, the performance of Domain-PFP is overall similar to the previous methods like DeepGraphGO and NetGO 2.0, while adding the information of BLAST and PPI substantially improved the performance. However, the code in GitHub provided by the authors seems to only support prediction using the base model (without BLAST and PPI). I suggest also include the “DPFP + BLAST + PPI” model.

4. Several typos should be fixed, and the names of the methods should be consistent in the whole paper:

a. “predictio” in Introduction line 7;

b. “ $p(\text{domain}|\text{GO})$ ” in Methods, Network training (should be $p(\text{GO}|\text{domain})$?)

c. “DomainGO-PFP prob” in section [GO function prediction by “DomainGO-PFP” in comparison with existing methods] (should be DomainGO-prob and Domain-PFP?)

Responses to Comments by Reviewer #1

Reviewer #1 (Remarks to the Author):

This paper presents Domain-PFP, a protein function prediction method using function-aware domain embeddings. The authors employ a self-supervised protocol to derive functionally consistent representations of domains by learning domain-Gene Ontology (GO) cooccurrences and associations. Specifically, the authors devise embeddings for domains so that functionally related domains have similar embeddings. The author carried out various experiments and discussed associations between Domain and GO and how Domain-GO associations learned by DomainGO-prob can be interpreted. And evaluations showed that protein representations using domain embeddings are superior to those of large-scale protein language models in GO prediction tasks. The authors also discuss the limitations of protein domain approaches: all existing protein domain-based methods fail to predict anything if the domain seen during inference does not exist in the training data. The authors suggest exploring larger deep-learning models to generate feature-aware domain representations. The following are some comments to improve the paper.

Major

1. NetGO 2.0 dataset used in this work is different from the one provided in the original paper of NetGO 2.0, which gives the url of google driver. The original data set should be used for evaluation.

>> We understand the concern of the reviewer. Actually, the NetGO2.0 benchmark dataset used in our work used the same file, which was provided in the google drive url (<https://drive.google.com/drive/folders/1wSS-R335UcNMToMskx3dE4XcTaLvCAOc>).

But we used the NetGO2.0 dataset that was split into training, validation, and test sets in the DeepGOZero paper by the Hoendorf lab, which is made available at (<https://deepgo.cbrc.kaust.edu.sa/data/deepgozero/data-netgo.tar.gz>). They made this split by following exactly the same procedure described in the NetGO2.0 paper. A practical issue of the NetGO2.0 benchmark dataset was that the data split into training, validation, and test set was only verbally described in their NetGO2.0 paper and the actual data splits were not provided. Therefore, we used the split data done by the Hoendorf lab.

Using the data split in the DeepGOZero paper was also convenient for comparison with other methods because they have compared DeepGOZero against existing methods, including DeepGOPlus, TALE+, and DeepGraphGO, and NetGO2.0 (server; they ran the web server) on the NetGO2.0 test set. The results of these methods we put in Table 1 are taken from the

DeepGOZero paper. DeepGOPlus, TALE+, and DeepGraphGO were trained on the training set of NetGO2.0 by the authors of DeepGOZero. Therefore, in Table 1, all the results from all the methods are tested on the same data from the NetGO2.0 dataset.

To clarify this, we added more explanation on page 20 in the main text and added details in Supplementary Table 3.

2. As mentioned in section “Comparison with large protein language models in GO function prediction”, a protein-level embedding contains 3 embeddings from MFO, BPO, and CCO, respectively. Why are there three representations of each protein? Are the three embeddings per domain trained from MFO, BPO, and CCO, respectively?

>> Yes, we trained three embeddings of the domains for the three sub-ontologies, following the common practice of training three different models for the three sub-ontologies. For example, DeepGOZero, TALE, DeepGraphGo, DeepGOPlus methods we compared in Table 1 have trained 3 separate models. Having an individual model for each of three categories would be able to capture graph structures of MFO, BPO, CCO, well, because these three graphs are not connected and each graph has different depths, nodes, and other graph features.

To clarify, we have explicitly mentioned this in the revised manuscript on page 6,

We trained three different versions of DomainGO-prob for the three sub-ontologies, MF, BP and CC, respectively.

Three different models, i.e., three different sets of embeddings were developed for the three sub-ontologies.

Also on page 16, we added “For this benchmark, we trained the domain and GO embeddings (Eq. 2) for the three GO categories separately..”

3. Different protein domains do not contribute equally to the final function of the protein. Compared with using the average method to integrate the embedding of the protein domain into the embedding of the protein, is it better to add weights?

>> This is a good point. We agree with the reviewer’s comment that not all protein domains contribute equally to the function. In addition to the potential different contribution of domains, studies have shown that the recurrence and order of domains also matter (Messih, M. A., Chitale, M., Bajic, V. B., Kihara, D., & Gao, X. (2012). Protein domain recurrence and order can

enhance prediction of protein functions. *Bioinformatics*, 28(18), i444–i450.
<https://doi.org/10.1093/bioinformatics/bts398>).

As a result, although average pooling is the most used approach to aggregate information in deep learning, it may not be sufficient to capture the complex contributions of the protein domains to the respective functions, as the reviewer mentioned. We have been thinking about a suitable modeling for this and we plan to use attention mechanism to put appropriate weight on the different domains, based on the context. We wish to explore this in our future work.

We added this as a future direction in Discussion (page 26).

4. For NetGO dataset and CAFA3 datasets, is the protein embeddings' dimension 256 or 768 when predicting a GO term?

>> For the NetGO and CAFA3 datasets, the protein embeddings are 256-dimensional. In fact, all the DomainGO-prob models trained by us used 256 as embedding dimension. It is mentioned in the Self-supervised learning for domain representation subsection (page 5) and Neural Network Architecture subsection in Methods (page 27).

Minor

1. There are some format issues, such as most equations are right-aligned, but some are centered.

>> Thank you for checking. We have revised the equations and made them all center-aligned.

2. In eq 6, 'GO Terms A' creates ambiguity and tends to make the readers think that it is a set related to GO Term A rather than a set related to domain A. It is suggested that the authors revise it.

>> We changed the notation to "*GO Terms_{domainA}*" and added the explanation on page 9.

3. Figure B in Figure 2 is difficult to understand. It is more appropriate to use the size of the circle to represent "Number of protein pairs", and use different colors to represent the "Std. funSim Score".

>> Thank you for your suggestion, we have revised Figure 2B.

4. The presentation form of Figure 5 is not very good. It is difficult to see the specific numerical differences at a glance, and such an X-axis is very inconvenient to read. Perhaps

you can swap the X and Y axes (Just like Figure 4. a). And the range of F-max on the axis can start not from 0, but higher (say 20).

>> Thank you for the suggestion, we have revised Fig.5.

Responses to Comments by Reviewer #2

Reviewer #2 (Remarks to the Author):

Comments to the authors

Because of the rapid increase in the number of proteins and the diversity of their functions, developing methods for the automatic and accurate prediction of protein functions has become an important issue. For this issue, the authors proposed their new function prediction method, Domain-PFP, which uses function-aware domain embedding representations of proteins, based on domain-GO association learned in a self-supervised manner. After they learned pseudo-labels of GO prediction probability from individual domain terms, they derived a representation of domains consistent with functional information to characterize protein sequences and used that representation to predict the protein functions. They found that such representations were functionally meaningful because their embedding distance showed correlation with functional similarity of GO terms, and showed that protein representations using their domain embeddings are superior to those of large-scale protein language models (PLMs) in GO prediction tasks, despite having only a fraction of the PLM complexity. Finally, they used a KNN classifier, with their learned embeddings, along with sequence similarity and PPI information, and demonstrated that their method exhibited competitive performance when compared with top-scoring methods in the CAFA3 evaluation. According to results reported by the authors, Domain-PFP seems effective, though their model has only a fraction of the PLM complexity.

Nowadays, structure prediction methods such as AlphaFold enable us to obtain more or less accurate structure models for most proteins, meaning that we can use them for structural information for large-scale protein function annotations. In fact, methods such as DeepFRI and GAT-GO have been developed. Consequently, including a systematic comparison with such methods is expected to be helpful for readers.

>> This is an important point. We thank the reviewer for pointing this. We have collected the dataset used by DeepFRI and GAT-GO and retrained Domain-PFP on the training dataset. The results are as follows:

Method	F _{max}			AUPR		
	MF	BP	CC	MF	BP	CC
Naive	0.156	0.244	0.318	0.075	0.131	0.158
BLAST	0.498	0.400	0.398	0.120	0.120	0.163
DeepGO	0.359	0.295	0.420	0.368	0.210	0.302
DeepFRI	0.542	0.425	0.424	0.313	0.159	0.193
GAT-GO	0.633	0.492	0.547	0.660	0.381	0.479
Domain-PFP	0.662	0.519	0.532	0.653	0.503	0.510
Domain-PFP + BLAST	0.746	0.611	0.605	0.754	0.606	0.612

The results of Naïve to GAT-GO are taken from the GAT-GO paper. It can be observed that Domain-PFP outperformed all the methods in all the metrics except for Fmax of CC and AUPR of MF, where GAT-GO had a higher values.

The performance of Domain-PFP was further improved by including BLAST information, which is not an unfair advantage as both DeepFRI and GAT-GO use sequence information through a protein language model (PLM).

We have added this new result as Supplementary Table 5 and discussed it on page 24.

In the manuscript, the authors stated that “We used self-supervised learning because it can directly learn domain and GO co-occurrence from abundant protein sequences and thus is able to alleviate the low-coverage problems of domain information”, but described that “We intended to keep the function f simple so that the domain embeddings could effectively learn functional relevance, rather than letting the function f learn the correlation between domain and GO term

co-occurrence". It seems that these might not be mutually consistent. We presume that the authors will discuss these points more carefully.

>>

In the first statement, we expressed that self-supervised learning has the potential to alleviate the issue of lack of function annotation to domains. Although domains have important function information of proteins, a practical problem is that the current domain database (e.g. InterPro) does not have function annotations for many of domains. Thus, in this work, by using self-supervised learning, which can compute a probability that a domain has a specific GO term, by analyzing domain and GO co-occurrence, we can alleviate the problem of the limitation of function annotation of domains.

On the other hand, in the second statement, we wanted to mention our design decision of keeping the function f simple. Our primary goal is to learn a functionally enriched embedding representation for the domains, i.e. designing a strong encoder for the domains. The function f can be treated as a decoder since we are trying to retrieve the domain-GO probability from the learned embedding. In recent deep learning, it has been demonstrated that a strong encoder in conjunction with weak decoder results in a strong representation learner (He, Kaiming, et al. "Masked autoencoders are scalable vision learners." *Proceedings of the IEEE/CVF conference on computer vision and pattern recognition. 2022.*). Therefore, we decided to make the f function as simple as possible to pass the heavy lifting to our embedding learner, expecting that the embedding will be a more meaningful representation.

We revised the sentences on page 3:

"We used self-supervised learning because it can directly learn domain and GO co-occurrence from abundant protein sequences and is able to alleviate the problem of current domain databases, where many domains do not have function annotation."

On page 6, we added a new sentence "This is inspired by a recent work, which demonstrated that a strong encoder in conjunction with weak decoder results in a strong representation learner." and cited the paper by He, Kaiming, et al. we mentioned above.

Responses to Comments by Reviewer #3

Reviewer #3 (Remarks to the Author):

This manuscript introduces a domain-based method to predict protein function(GO terms), which differs from the previous domain-based methods mainly in using a pre-training step to learn the domain embeddings by modeling the domain-GO cooccurrences and associations. The overall

framework is interesting, and the paper is well written. However, some details in the experiments are not very clear, and the discussions of the latest methods should be updated to be more comprehensive. Below are some specific comments that the authors may find useful.

1. When comparing Domain-PFP with other methods in the NetGO 2.0 and CAFA3 datasets, did the authors use the training sets to pre-train DomainGO-prob? Please add more details explicitly. If not, did the authors use all annotations in Swiss-Prot for pre-training? This will lead to the data leakage problem. If yes, I don't think using the GO annotations from the training sets to learn GO-domain associations can be claimed as "self-supervised". I suggest the authors rephrase some of the sentences.

>> We agree with the reviewer that using all the annotations from Swiss-Prot during pre-training will result in data leakage as the test proteins of these benchmarks can likely be used in the model training, which would lead to an unfair comparison. Therefore, we trained Domain-PFP on the training set provided in the NetGO2.0 and CAFA3 datasets, respectively.

We clarified this on Page 20 for the NetGO benchmark:

We trained DomainGO-prob on the NetGO training dataset ...

For the CAFA3 benchmark, we mentioned this on Page 24 similarly:

We trained the network model of Domain-PFP using the CAFA3 training dataset and evaluated the results using the official evaluation code.

Regarding the other comment on presenting our learning algorithm as a self-supervised learning (SSL), we would like to mention that there are a number of variants of self-supervised learning. Our method follows one of the core concepts of SSL where pseudo-labels are first learned to initialize model parameters, which are then used to perform the actual task using a supervised or unsupervised method (Doersch, Carl; Zisserman, Andrew (October 2017). "Multi-task Self-Supervised Visual Learning". *2017 IEEE International Conference on Computer Vision (ICCV)*. IEEE. pp. 2070–2079.). (This definition is also presented in Wikipedia). Our approach is consistent with this definition as we first use the domain-go association probabilities as pseudo labels, which initializes our domain embedding parameters; then, we use this embedding later in a supervised learning protocol and we predict the functions of the proteins. This strategy also hold in the benchmarks on the NetGO and CAFA3 dataset.

To clarify what we mean by self-supervise learning, we cited the above paper by Doersch et al. and discussed it in Discussion (page 26).

2. Please update the introduction or comparison to include more latest methods, e.g., language-model-based function predictor SPROF-GO [doi 10.1093/bib/bbad117].

>> Thank you for referring the paper. The SPROF-GO method is quite recent one and it was published after our manuscript had been written. We now cited the SPROF-GO paper in the introduction section where we discussed about protein-language models (page 4).

3. As shown in Table 1 and Figure 5, the performance of Domain-PFP is overall similar to the previous methods like DeepGraphGO and NetGO 2.0, while adding the information of BLAST and PPI substantially improved the performance. However, the code in GitHub provided by the authors seems to only support prediction using the base model (without BLAST and PPI). I suggest also include the “DPFP + BLAST + PPI” model.

>> We thank the reviewer for making this suggestion, which will certainly improve the usability of our released implementation of Domain-PFP. We have now added the option of including BLAST and PPI information in the prediction script. The users can add optional flags, e.g., --blast and/or --ppi in the command to use the respective additional information. For cases, where no blast or ppi match is found, they are just discarded and the user is notified of this. In addition, we have incorporated this in our google colab notebook.

4. Several typos should be fixed, and the names of the methods should be consistent in the whole paper:

- a. “predictio” in Introduction line 7;
- b. “p(domain|GO)” in Methods, Network training (should be p(GO|domain)?)
- c. “DomainGO-PFP prob” in section [GO function prediction by “DomainGO-PFP” in comparison with existing methods] (should be DomainGO-prob and Domain-PFP?)

>> Thank you for catching these typos. We have fixed them.

Reviewers' comments:

Reviewer #1 (Remarks to the Author):

The author has made comprehensive and complete revisions to the article based on the reviewers' comments. Specifically, the author added DeepFRI and GAT-GO as Baseline to conduct more complete comparative experiments, and additionally discussed the latest method SPROF-GO. In addition, the author updated the github code to include the "DPFP + BLAST + PPI" model. Overall, I think the vast majority of comments have been well addressed. Here are some minor issues.

(1) The data split of NetGO 2.0 was actually provided in the google driver. The authors are suggested to include the results of Domain-PFP in the supplementary materials over the original NetGO 2.0 dataset.

(2) The alignment of the Figure 3 images looks a little weird. They appear to be neither left nor center aligned.

Reviewer #2 (Remarks to the Author):

The authors have addressed all my concerns. Nevertheless, there is a tiny question. Did the authors show the results of Naïve to GAT-GO, instead of DeepFRI, from the GAT-GO paper? It seems that the results of Naïve to GAT-GO shown in Supplementary Table 5 are the same with the ones in the GAT-GO paper. Otherwise, the authors obtained the same result for GAT-GO by themselves?

Reviewer #3 (Remarks to the Author):

The authors have addressed all my comments.

Reviewer #1 (Remarks to the Author):

The author has made comprehensive and complete revisions to the article based on the reviewers' comments. Specifically, the author added DeepFRI and GAT-GO as Baseline to conduct more complete comparative experiments, and additionally discussed the latest method SPROF-GO. In addition, the author updated the github code to include the "DPFP + BLAST + PPI" model. Overall, I think the vast majority of comments have been well addressed. Here are some minor

issues.

>>> We thank the reviewer for the constructive and insightful comments.

(1) The data split of NetGO 2.0 was actually provided in the google driver. The authors are suggested to include the results of Domain-PFP in the supplementary materials over the original NetGO 2.0 dataset.

>>> We apologize for misinterpreting the reviewer's initial comment. We have evaluated Domain-PFP on the train-validation-test split used in the original NetGO benchmark. The results have been included as Table 6 in the supplementary materials and mentioned in page 24. We copy the table below.

Method	F _{max}			AUPR		
	MF	BP	CC	MF	BP	CC
Naive	0.416	0.256	0.542	0.276	0.118	0.464
BLAST-KNN	0.632	0.312	0.566	0.542	0.132	0.405
LR-3mer	0.427	0.258	0.552	0.317	0.125	0.478
LR-InterPro	0.651	0.325	0.641	0.623	0.166	0.587
Net-KNN	0.519	0.325	0.596	0.416	0.192	0.528
RNN	0.524	0.265	0.574	0.424	0.124	0.477
LR-Text	0.464	0.248	0.479	0.353	0.154	0.403
DeepGOPlus	0.620	0.305	0.620	0.521	0.115	0.493
GOLabeler	0.667	0.326	0.631	0.647	0.193	0.557
NetGO	0.674	0.362	0.646	0.653	0.239	0.583
NetGO 2.0	0.666	0.366	0.663	0.655	0.269	0.593

Domain-PFP	0.668	0.382	0.672	0.666	0.322	0.689
Domain-PFP + BLAST	0.668	0.407	0.677	0.688	0.344	0.709
Domain-PFP + PPI	0.651	0.409	0.667	0.672	0.353	0.664
Domain-PFP + BLAST + PPI	0.675	0.425	0.678	0.708	0.365	0.676

We evaluated Domain-PFP on the same train-validation-test split used in the original NetGO benchmark. Results of Naive to NetGO2.0 were taken from the NetGO2.0 paper. The top three scores for each of the metrics are highlighted by bold font, double underline, and single underline, respectively.

(2) The alignment of the Figure 3 images looks a little weird. They appear to be neither left nor center aligned.

>>> We thank the reviewer for pointing this out. We have revised the figure and made it left aligned.

Reviewer #2 (Remarks to the Author):

The authors have addressed all my concerns. Nevertheless, there is a tiny question. Did the authors show the results of Naive to GAT-GO, instead of DeepFRI, from the GAT-GO paper? It seems that the results of Naive to GAT-GO shown in Supplementary Table 5 are the same with the ones in the GAT-GO paper. Otherwise, the authors obtained the same result for GAT-GO by themselves?

>>> We apologize for making the confusion, this was a typo. We actually obtained the results of Naive to GAT-GO from the GAT-GO papers, we have corrected this in the revised supplementary material.

Reviewer #3 (Remarks to the Author):

The authors have addressed all my comments.

>>> We thank the reviewer for the comments and suggestions.